# Super Shedding in Enteric Pathogens: A Review

**DOI:** 10.3390/microorganisms10112101

**Published:** 2022-10-22

**Authors:** Florent Kempf, Roberto La Ragione, Barbara Chirullo, Catherine Schouler, Philippe Velge

**Affiliations:** 1INRAE, Université de Tours, ISP, F-37380 Nouzilly, France; 2School of Veterinary Medicine, Faculty of Health and Medical Sciences, University of Surrey, Guildford GU2 7XH, UK; 3School of Biosciences and Medicine, Faculty of Health and Medical Sciences, University of Surrey, Guildford GU2 7XH, UK; 4Department of Food Safety, Nutrition and Veterinary Public Health, Istituto Superiore di Sanità, Viale Regina Elena 299, 000161 Rome, Italy

**Keywords:** super shedding, enteric pathogens, gut microbiota, host response, bacterial polymorphism, control strategies

## Abstract

Super shedding occurs when a small number of individuals from a given host population shed high levels of a pathogen. Beyond this general definition, various interpretations of the shedding patterns have been proposed to identify super shedders, leading to the description of the super shedding phenomenon in a wide range of pathogens, in particular enteric pathogens, which are of considerable interest. Several underlying mechanisms may explain this observation, including factors related to the environment, the gut microbiota, the pathogen itself (i.e., genetic polymorphism), and the host (including immune factors). Moreover, data suggest that the interplay of these parameters, in particular at the host–pathogen–gut microbiota interface, is of crucial importance for the determination of the super shedding phenotype in enteric pathogens. As a phenomenon playing an important role in the epidemics of enteric diseases, the evidence of super shedding has highlighted the need to develop various control strategies.

## 1. Introduction

Heterogeneity in infectious disease dynamics, whereby a small subset of individuals is involved in the majority of the transmission events, has been described for several pathogens. These include viruses such as influenza [1], MERS (Middle East respiratory syndrome-related coronavirus; [2]), SARS (severe acute respiratory syndrome coronavirus), and SARS-CoV-2 [3], as well as several bacterial pathogens, including *Escherichia coli* and especially *E. coli* O157:H7 [4], *Coxiella burnetti* [5], *Klebsiella pneumoniae* [6], *Salmonella* [7], and *Campylobacter* [8]. Super shedding appears as a particular feature of heterogeneous infection, in that it should be distinguished from shedding patterns differing among host species. Super shedding occurs when a small number of a given host population shed high levels of a pathogen as a result of a successful infection (which implies successful survival, colonization, and persistence). Super shedding is of particular interest with respect to the development and implementation of control strategies. Indeed, numerous analyses have described that 20% of animals (super shedders) are responsible for 80% of cross contaminations between animals [9]. Moreover, control programs that fail to reach the super shedder group may be less effective than expected [10]. The purpose of this review is to analyze the current knowledge available on super shedding of bacterial pathogens of medical and veterinary interest, specifically enteric pathogenic bacteria. As the mechanisms have been described in mice and farm animals, in this review, we will mainly focus on animal models. Finally, we will explore the issues and implications related to this phenomenon with respect to pathogen control.

## 2. What Is a Super Shedder?

### 2.1. Super Shedding and Infectiousness

The phenomenon of a small proportion of host individuals showing a high level of pathogen shedding is known as super shedding. A significant effort has been made to distinguish the concept of super shedding from that of super spreading. Chase-Topping [4] suggested that a super shedder should be considered an individual that can shed more pathogens than other host individuals, whereas a super spreader has more opportunities to infect other hosts with a given pathogen type than most other individuals of the same host type. Both phenomena may lead to heterogeneity of infection and transmission, and the super shedding may lead to super spreading. However, they are independent traits; a super shedding mostly corresponds to the host–pathogen relationship, whereas super spreading corresponds to host-to-host interactions, taking into account the environment [4]. In this case, the probability that the pathogen can survive in the environment and the distance between contaminated and naïve hosts also play a role in super spreading.

Epidemiologic modelling of pathogen transmission has shed new light on both concepts. The reproduction number (R0) reflects the ability of a pathogen to spread in a host population. It corresponds to the number of secondary infections produced when one infected individual is introduced into a host population in which all individuals are susceptible [11]. This population average does not account for the variation in the number of secondary cases and therefore does not reflect the role of individual variation in outbreak dynamics [12]. For this reason, Lloyd Smith et al. [13] introduced an ‘individual reproduction number’, defined as the expected number of secondary cases caused by a particular infected individual. At the population level, the individual reproduction number is distributed around the mean R0; super spreaders are realizations from the right-hand tail of the R0 distribution. This concept unified the theories of super shedding and super spreading; according to VanderWaal and Ezenwa [12], individual reproduction encompasses both physiological (donor infectiousness and receiver susceptibility) and behavioral components of transmission (contact rate). Super spreading corresponds to a maximization of the host- and behavior-related parameters, whereas super shedding corresponds to a high level of infectiousness.

The fact that super shedders are not necessarily super spreaders is known as the “super shedder paradox”, whereby high shedding levels may have little impact on infection dynamics [14,15]. Explanations for this apparent paradox include that (1) the direct contact among infectious and susceptible animals are required for the transmission, (2) the transmission is not linear with respect to the absolute number of pathogens present in the environment [16], and (3) environmental survival of the pathogen may vary (intra-individual and intra-bacteria variabilities).

### 2.2. Super Shedders as a Distinct Category: The 80/20 Rule

Super shedders can be considered a distinct individual host category. Accordingly, field studies have revealed that a few individuals may be the main drivers of pathogen infection. For instance, in *Escherichia coli* O157:H7, Omisakin et al. [17] observed that 4 of 44 animals under scrutiny shed more than 96% of the total number of pathogens. It has been suggested that a robust picture arises from observational and modelling studies; 80% of transmission is caused by 20% of the most infectious animals. The so-called ‘80/20 rule’, initially proposed for sexually transmitted and vector-borne diseases [10], was later generalized to a larger variety of pathogens [9]. This categorization has strong implications for control strategies because interventions targeting the most infectious individuals may thus contribute to a significant reduction in the reproduction number (R0) [9,18].

### 2.3. Temporal Shedding Patterns

Defining a super shedder may be challenging due to the contrasting intra-individual shedding patterns. For example, shedding may be variable over time in the same animal. Intermittent shedding has been reported for *Salmonella* Typhi [19], *E. coli* O157:H7 [20], and *Mycobacterium avium* subspecies *paratuberculosis* [21]. For *S.* Typhi, intermittent shedding in humans is commonly referred to as ‘showers of *S.* Typhi’. In *E. coli* O157:H7, short-term studies have demonstrated that bacteria are shed intermittently in most animals [20,22]. In their study, Robinson et al. [22] showed that within-individual shedding variation was higher than the between-individual variation. These individuals shed bacteria for durations ranging from 4 days [20] to 2 weeks [22]. In *M. avium* subspecies *paratuberculosis*, intermittent shedders refer to naturally infected animals that may switch between a low and no shedding state; they belong to a category distinct from super shedders [21]. Because of these contrasted individual shedding patterns, some authors have proposed that super shedding does not form a distinct category of individuals but that it should be considered as a transient property that appears and disappears over time [14,23]. Moreover, it has been suggested that super shedding may be an outcome of environmental factors rather than the effect of intrinsic host factors [24]. However, some observations may support the relevance of the super shedder category in certain models; for instance, some causal factors, including antibiotic treatment and gut microbiota composition, may be related to a persistent *Salmonella* shedding phenotype [7,25]. Moreover, intermittent and persistent shedding may depend on the method used to collect samples. For example, Velge et al. [26] reported that cloacal swabs often indicate the intermittent presence of *Salmonella*, contrary to fresh fecal samples. Nevertheless, intermittent shedding remains an important issue with respect to disease control; mathematical modelling of disease transmission that neglects intermittent infectiousness may lead to biased estimates of the reproduction number (R0) and infection prevalence at the time of slaughter [27].

### 2.4. Delineation of the Super Shedding Category

The delineation of super shedder category stems from the characterization of shedding patterns, which can be assessed using several quantitative methods (in practice, most studies are based on pathogen counts) and sampling method (swabbing, collection of cecal or fecal material from dead or living hosts). To take into account the facets of the super shedding phenomenon, several methods have been proposed to delineate a super shedder category in a given population. Depending on the host and pathogen under scrutiny, and on the method of sampling, different absolute thresholds have been proposed. For instance, for *Salmonella* Typhimurium, Lawley et al. [7] proposed that mice shedding more than 10^8^ CFU/g feces at three consecutive time points should be considered super shedders (in their study, fecal shedding was monitored every 1 to 3 days). For *M. avium* subspecies *paratuberculosis*, super shedder cows are classically considered those excreting more than 10^4^ CFU/g feces [28]. For *E. coli* O157:H7, the values found in the literature include 10^4^ CFU/g feces [17,29] and 10^3^ CFU g/feces [14,22]. This threshold-based approach results from a mixture distribution analysis conducted by Chase-Topping et al. [30] but does not take into account the variations in excretion levels often observed with age or post-infection duration for certain super shedding phenomenon. Cobbold et al. [31] proposed a different approach for *S.* Typhimurium also based on a threshold and on temporal dynamics. Individuals were defined as super shedders on the basis of both high pathogen mean (10^4^ CFU/RAMS for recto-anal mucosal swabs) and persistent colonization (four consecutive positive samples). Beside the threshold-based approach, some authors have proposed alternative methods exploiting temporal dynamics (Figure 1). First, temporal patterns may be described by the area under the plotted log curve (AULC) (Figure 1A). Under this approach, the fecal counts at different time points are log-transformed; the shape of the obtained curve is summarized by computing the area under the curve. This approach allows for accurate comparison of curves presenting with a similar shape (e.g., constant growth of the shedding levels) but does not distinguish between curves presenting the same AULC with different kinetics (e.g., constant increase or decrease in the shedding levels) (e.g., [32]). Second, the shedding levels may be summarized by hierarchical clustering of the values obtained at different time points (Figure 1B,C). Under this approach, decreasing and increasing shedding patterns are clustered separately, and the super shedding category is based on the highest mean level of bacteria [33]. In a similar fashion, the shedding levels can be clustered by principal component analysis [25] (Figure 1D,E).

## 3. Enteric Bacteria and Super Shedding

Heterogeneity of infection is a multifactorial process involving pathogen features, dose, route of infection, feed, environment, and variability of host genetics. However, there is growing evidence that the gut microbiota plays a role in the development of the super shedding phenomenon, in particular in enteric bacteria, which can be implicated in a broad range of diseases of major economic, veterinary, and medical interest.

### 3.1. Escherichia coli

Pathogenic *E. coli* include those causing enteric diseases that have been divided into fifteen main pathotypes [37]. Among them, Shiga toxin-producing *E. coli* (STEC, also referred to as enterohemorrhagic *E. coli* (EHEC)) are a serious threat to public health around the world (7775 cases reported at the EU level in 2019, representing the third most frequent bacterial agent detected in food-borne outbreaks, EFSA 2021). STEC include more than 100 serotypes, with *E. coli* O157 and O26 being the most commonly reported in human cases [38]. Infection by *E. coli* O157 may be associated with a wide range of symptoms, including severe hemorrhagic diarrhea. It can progress to more serious illness, including hemolytic uremic syndrome (HUS) and death [4]. Cattle are considered the main reservoir host and are healthy carriers [39]. Super shedding has been described in cattle infected with *E. coli* O157:H7, with longitudinal studies demonstrating that shedding is not detected in the majority of cattle groups, although some groups include high proportions of super shedders [9]. O157:H7 super shedding was partially reproduced in an experimental study [40]. O157:H7 super shedding has been also reported in other animal hosts, such as sheep [41] and chicken in a strain-dependent fashion [42]. In this serotype, super shedding has been also reported for *E. coli* O26 infecting cattle [43]. A few studies also reported that low and super shedders may be observed in pigs infected with another pathotype, enterotoxigenic *E. coli* (pathotype ETEC; [44,45]).

### 3.2. Salmonella

*Salmonella* are enteric bacteria of major economic and sanitary interest worldwide. Salmonellosis is the second most commonly reported gastrointestinal infection in humans after campylobacteriosis and an important cause of food-borne outbreaks (e.g., in the European Union, 87,921 confirmed cases of salmonellosis were reported in 2019) [38]. *Salmonella* infections cause significant losses in the agri-food industry and represent a burden for healthcare systems. Depending on the hosts and serotypes, *Salmonella* can induce diseases ranging from gastroenteritis to typhoid fever. Typhoid fever is caused by the human-associated serovar *S.* Typhi, which causes 16 million new cases and more than 600,000 deaths per year [46]. The bacteria may induce a carrier state, whereby the bacteria persist asymptomatically for many weeks up to many years. In *Salmonella* Typhi, carrier host individuals may temporally become *Salmonella* super shedders. Two well-documented cases of *S.* Typhi human carrier individuals have highlighted the relevance of super shedding for public health systems (i.e., ‘Typhoid Mary’ and ‘Mr. N’ [19]). The two main non-typhoidal serovars involved in human infections are *S.* Typhimurium and *S.* Enteritidis, which are also able to induce symptomatic and asymptomatic infections in pigs and poultry [38]. *Salmonella* carrier animals, especially super shedders, also represent a serious safety issue due to the contamination of food products and asymptomatic transmission within the flock or the farm. To date, animal hosts in which the *Salmonella* super shedding phenomenon has been observed through experimental studies include mice [7], chickens [33], cattle [18], and pigs [34].

### 3.3. Mycobacterium avium subspecies paratuberculosis

*Mycobacterium avium* subspecies *paratuberculosis* (MAP) is the causative agent of an important mycobacterial infection primarily associated with domestic ruminants but has also been isolated in a wide range of non-ruminant animals, including humans [47]. Its economic impact has been estimated to be more than USD 200 million per year in the United States [48]. MAP may also be a threat for human health, insofar as it has been associated with an increased risk of developing Crohn’s disease [47]. MAP is a slow-growing bacterium; the infection shows a slow progression from the initial state to clinical disease. Infected cattle are categorized in one of three stages, depending on fecal shedding and clinical signs. In stage 1, no bacteria are detectable by culture, and the animals are still preclinical. In stage 2, cattle shed MAP without clinical signs. Lastly, in stage 3, the culture becomes positive, and symptoms of so-called Johne’s disease arise (weight loss, diarrhea) [49]. Only few animals develop clinical symptoms. Empiric observations of heterogeneous MAP shedding levels have led to the definition of three categories: low, moderate, and high shedders [50]. Despite significant differences reported in field studies, super shedding has also been also observed following experimental infections [21]. On one hand, for a few individuals, natural infections involve a long period with no shedding, followed by a short period of high shedding. On the other hand, experimental infections lead to much more complex shedding patterns with rapid progression and significant fluctuations in shedding levels.

### 3.4. Campylobacter spp.

Campylobacteriosis was the most commonly reported gastrointestinal infection in humans in the EU in 2019 (220,682 human cases [38]). Although this category corresponds to all infectious diseases caused by the genus *Campylobacter*, the only form of campylobacteriosis of major public health importance is *Campylobacter* enteritis due to *C. jejuni* and *C. coli.* In humans, the clinical symptoms of campylobacteriosis range from watery, non-bloody, non-inflammatory diarrhea to severe inflammatory diarrhea with abdominal pain and fever [51]. It can be found in almost all animals of economic interest. For example, Campylobacter infections in poultry are usually asymptomatic, except at a young age [52]. Observations consistent with the existence of *Campylobacter* super shedding have been reported in broiler chickens [8,53]. In line with this, variation in shedding patterns has been observed in *Campylobacter*-infected cattle [54].

## 4. What Causes Super Shedding?

### 4.1. Environmental Factors

Several studies have shown that environmental factor may determine the super shedding phenomenon. The seasonality of *E. coli* O157:H7 shedding is widely reported [55], with increased shedding levels during summer months. For instance, Ogden et al. [41] reported a reduced prevalence in summer but more intense shedding. Other studies reported an increase prevalence of *E. coli* O157:H7 shedding in summer [55], suggesting summer conditions encompass environmental factors that favor super shedding. This includes extrinsic factors favoring pathogen survival in the environment, thus causing increased exposure. Modification of host intrinsic factors by environmental factors, such as animal predisposition or gut microbial composition, should also be taken into account.

These patterns of seasonality are not confirmed by other studies. For example, in a longitudinal study including 52 dairy heifers. Williams et al. [56] observed a decrease in super shedding at high temperatures and under high solar exposure. This result may be explained by the small sample size or the major effect of solar exposure on pathogen survival when temperatures are very high. Other analyses of climate factors pointed a link between *E. coli* O157:H7 super shedding, rainfall, and relative humidity [56], which could participate in the contamination of leafy vegetables.

Intrinsic seasonal effects that could play a role in peak shedding during the summer months include livestock feed [55] or a predisposition caused by the first lactation period [57]. In addition, using an experimental design, Edrington et al. [58] found a positive correlation between prevalence and day length. The authors suggested an impact of melatonin secretion on the immune system to explain this correlation.

Increased *E. coli* O157:H7 shedding in summer may be related to the interplay between intrinsic and extrinsic factors. Surprisingly, in their experimental study attempting to reproduce the link between shedding and summer conditions (including temperature, daylight, and humidity fluctuations), Sheng et al. [59] found that winter conditions were more prone to favor super shedding. According to the authors, this may reflect the fact that the increased shedding naturally observed during summer may be caused by extrinsic factors rather than intrinsic factors.

Among non-seasonal factors, hide cleanliness may be negatively associated with *E. coli* O157:H7 super shedding [56,60]. Dirtiness may be related to animal behavior (recurrent licking) and micro-environmental conditions (presence of feces, humidity) that favor high levels of *E. coli* O157:H7 and subsequent recontaminations. Fecal consistency was also found to be related to super shedding, which may reflect an impact on the gastrointestinal tract [56]. Analyses focusing on husbandry practices also found factors favoring *E. coli* O157:H7 shedding, including the spread of slurry rather than manure, the use of pasture rather than pens, change in diets [61] and pasture growth [56], and movement of female breeding cattle and weaning [30]. The role of diet composition, in particular, has been extensively reviewed elsewhere (e.g., [62]).

Despite numerous studies on *E. coli* O157:H7 and environmental causes of super shedding, only a few studies have addressed this question in other models of super shedding. Crossley et al. [49] observed more *M. avium* subsp. *paratuberculosis* high shedders in winter. However, this observation may be explained by husbandry practices rather than by a true biological phenomenon. In an experimental challenge study based on *Salmonella*-infected chickens, Traub-Dargatz et al. [63] did not observe any link between heat stress and fecal shedding.

### 4.2. Gut Microbiota-Related Factors

In a pioneering paper, using a mouse model, Lawley et al. [7] demonstrated that indigenous intestinal microbiota controls the development of the super shedder phenotype. Recent progress has been made in identifying microbiota–pathogen interactions involved in the pathogen colonization phenomenon [64]. Two main hypotheses can be made concerning the link between the gut microbiota composition and the occurrence of super shedding.

First, low diversity may favor colonization of the pathogen due to increased niche availability and reduced colonization resistance, suggesting that the super shedder phenotype could be linked with low α diversity, whereas the low shedder phenotype one could be linked with high α diversity (assuming the correct use of the α-diversity index correlating high values and high diversity; e.g., Shannon index). Consistent with this idea, using a model of weaned pigs naturally infected with *Salmonella*, Argüello et al. [65] observed that the establishment of a diverse and healthy microbiota may hamper the colonization success of *Salmonella*, although the level of *Salmonella* shedding was not related to the α diversity. In a study based on chickens challenged at 2 days of age with *Salmonella*, Pedroso et al. [66] observed increased α diversity and reduced cecal colonization after infection. Similarly, in field studies, Zhao et al. [67] reported increased diversity in *E. coli* O157:H7 non-shedder animals, as similarly reported by Stenkamp-Strahm et al. [68]. However, the α diversity before infection may not solely explain the shedding phenotype; when considering *E. coli* O157:H7 in field studies, Xu et al. [69] and Zaheer et al. [70] found that super shedder animals exhibit increased α diversity. In contrast, Bibbal et al. [71] did not observe α-diversity differences between shedder and non shedder animals. In *Campylobacter jejuni*-shedding animals, Sofka et al. [72] found a link between an increased α-diversity and the occurrence of shedding. Finally, Kaevska et al. [73] observed increased α-diversity dairy cows that not shed *M.* avium subsp. *paratuberculosis*. Given these conflicting results, it seems difficult to support the hypothesis that gut microbiota diversity plays a determining role in the super shedder phenotype. An interesting result is the observation of increased α diversity in super shedder chickens, which is correlated with the presence/absence of some bacterial genera in the super and low shedder phenotypes both before and after infection [25].

A second hypothesis is that the presence of specific features of the gut microbiota inhibits (directly of via byproducts) the growth of the pathogen, causing a low shedder phenotype. For instance, it has been suggested that short-chain fatty acids (SCFAs) may reduce colonization by *Enterobacterales* [64]. In line with this, Argüello et al. [65] obtained a few differential taxa abundances in *Salmonella* non-shedder pigs, including an enrichment of OTUs assigned to the families *Lachnospiraceae* and *Ruminococcaceae*, two major butyrate producers. In addition, using pig experimental infections, Bearson et al. [32] observed differences between low and super shedders of *Salmonella,* including a reduced abundance of *Ruminococcaceae* in super shedders before *Salmonella* challenge and, after infection, an overall increase in α diversity driven by the decrease in *Prevotella* and subsequent increases in several genera. *Prevotella*, a propionate producer, has been associated with *E. coli* O157:H7 low shedders. For instance, Zaheer et al. [70] observed an enrichment of *Prevotella* in the gut of *E. coli* O157:H7 non-shedders, in addition to *Treponema*. In their study based on the same animals, Wang et al. [74] observed another set of differences by considering the mucus of the rectal-anal junction as a separate compartment. They observed two OTUs enriched in super shedders (most likely assigned to *Bacteroides* and *Clostridium*) and seven OTUs enriched in nonshedders (*Coprococcus*, *Prevotella*, *Clostridium*, *Paludibacter*, and Proteobacteria). However, the presence of SCFA producers cannot systematically explain the emergence of super shedders; for example, Xu et al. [69] reported several features differentiating *E. coli* O157:H7 low and super shedders; among the 72 differentially abundant OTUs, one *Blautia*, one *Oscillospora*, three *Clostridium*, seven *Prevotella*, and one *Alistipes* OTU were found to be more abundant super shedders. Zaheer et al. [70] observed several genera with increased abundance in the gut of super shedders (*Ruminococcocus*, *Selenomonas*, *Campylobacter*, and *Streptococcus*). Using a model of *S.* Enteritidis heterogeneous shedding, Wu et al. [75] demonstrated that chicks presenting the lowest shedding levels harbored a gut microbiota enriched with *Desulfovibrio piger* post infection, whereas those presenting with the highest shedding levels were enriched with *Bacteroides caecicola*, a propionate producer, and *Helicobacter pullorum*.

SCFA-driven suppression of pathogen growth is not the only mechanism reported by Rogers et al. [64]; depletions of critical resources, such as crucial amino acids, may be included. An increasing number of articles describe how some commensal bacteria can protect hosts from infection and benefit their health. However, microbiota compounds or changes in the intestinal landscape induced by gut microbiota have the potential to facilitate colonization by pathogens and pathobionts [76].

After the establishment of an inflammatory response, other specific features of the gut microbiota features can enhance the growth of the pathogen [64]. Under these conditions, a new niche emerges that is favorable to the Enterobacterales over anaerobic bacteria. The bacteria closely related to the pathogen, such as Enterobacterales in the case of *Salmonella*, may play a crucially important role in the competition for the new emerging niche. Consistent with this idea, Kempf et al. [25] demonstrated that an early inoculation of a bacterial mix including *E. coli* strain Nissle 1917 reduced *Salmonella* shedding levels.

Taken as whole, the abovementioned gut studies show a large variety of taxa that may be involved in the emergence of low or super shedders. The lack of overlap among the observations made in the different studies highlight the need for further investigations. Although they may suggest convincing mechanistic scenarios, the correlation between the presence of a pathogen at high levels and other features do not allow for conclusions about the link of causality (e.g., whether the presence of the pathogen influences the gut microbial composition or if shedding levels depend on the gut microbial composition). Generally, the correlations between gut microbial features and super shedding highlight that different gut compositions may be related to the same shedding phenotype. The fact that different taxonomic features may encompass the same functions at the scale of the whole gut microbiota may explain these observations. Much remains to be investigated with respect to the functions involved in the super shedding phenomenon. For instance, Wang et al. [74] found 3 pathways with increased abundances in *E. coli* O157:H7 super shedders, including gene families associated with the biosynthesis of bacterial antimicrobial compounds targeting Gram-positive bacteria, and 12 with reduced abundances in super shedders, including gene families associated with LPS biosynthesis. This result likely reflects the comparatively lower abundance of Gram-negative and inhibitory Gram-positive bacteria in *E. coli* O157:H7 super shedders. In addition, microbial nutrient metabolism by the local microbiota may be associated with shedding patterns.

Other features of the gut microbiota should also be considered. For instance, Hallewell et al. [77] reported differing bacteriophage abundances in *E. coli* O157:H7 low and super shedders. In particular, phages presenting higher lytic capabilities were more frequently isolated in low shedders, suggesting that they may contribute to the mitigation of *E. coli* O157:H7 shedding. Owing to the lack of dedicated studies, the impact of phages may be underestimated; they can nevertheless account for as much as 80% of total bacterial mortality in a given ecosystem, considerably affecting bacterial diversity and functions [78]. This highlights the need to conduct more virome studies.

### 4.3. Bacterial Genetics/Polymorphism

Bacterial pathogens have evolved a wide range of strategies to colonize and invade the organs of their host, despite the presence of multiple host defense mechanisms. Compared to opportunistic bacteria, pathogens have the capability to overcome the host’s colonization resistance [64]. Virulence factors can determine several changes in the host’s intestinal epithelium, which in turn favor the growth of the pathogen. An open question is whether some bacterial strains induce more shedding than closely related strains. Different approaches (implying different definitions of a super shedder strain) may be undertaken. First, mutants targeted for key functions of the super shedding phenotype can be considered (e.g., [7]). Second, comparative studies can be conducted with strains either isolated from low and super shedder hosts either naturally (e.g., [79]) or experimentally (e.g., Kempf et al. personal data) infected.

The first approach may reveal specific features of the super shedder mutant strains. In line with this, in their mouse model, Lawley et al. [7] found that the *Salmonella* super shedder phenotype requires the presence of two determinants, SPI1 and SPI2 pathogenicity islands, harboring virulence factors T3SS-1 and T3SS-2. Nevertheless, the same strain may lead either to a super shedder or to a low shedder phenotype. Lawley et al. [7] did not observe heritable differences between super and low shedder *S.* Typhimurium strains.

Comparative studies falling into the second category sometimes do not reveal differences between low and super shedder strains. For instance, Munns et al. [79] did not find SNPs providing genetic segregation of *E. coli* O157:H7 low and super shedder strains. They also suggested the potential role of genes with unknown function. These results also suggest that there is no or few genomic differences at the strain level between the phenotypes. Similar conclusions have been drawn for *Salmonella* (Kempf et al. personal data). In these models, the role of other factors (e.g., related to the gut microbiota or the host) may intervene jointly and be more important than strain-level differences with respect to the emergence of the super and low shedder phenotypes.

However, in other cases, comparison of super shedder and control strains revealed that they differ at the genomic level. The specificity of the super shedder strains mostly lies in SNP polymorphism, prophage-associated genes, and extrachromosomal elements [80,81]. For instance, Cote et al. [81] reported between 310 and 4847 SNP differences among the super shedder strain and other O157:H7 strains under scrutiny, mostly included in virulence genes involved in adherence. Katani et al. [80] observed a number of SNPs, ranging between 384 and 3106 between O157:H7 super and low shedder strains; in addition, the super shedder strains under scrutiny differed in terms of the presence of a particular plasmid, specific phage insertion patterns, and SNP polymorphism. Another study suggested an association between phage type of the O157:H7 strain and shedding levels [30]; in this case, the strain-level differences may be related to the expression of genes harbored by inserted bacteriophages, in particular the genes located in the O island-phage repertoire [4]. O islands thus contain effector proteins implied in host tissue colonization. In line with this, Cote et al. [81] reported that phage insertion patterns may be a major feature of the recent evolution of super shedder strains. These results highlight the need for large-scale genomic comparisons including closely related super shedder strains.

In some cases, differences have been observed at the level of the bacterial metabolism. In line with this, Munns et al. [79] observed differences in the metabolism of several substances in *E. coli* O157:H7 super shedder strains, including an increased utilization of galactitol, thymidine, and 3-O-β-D-galactopyranosyl-D-arabinose. The differences related to bacterial metabolism may fall into a broad definition of virulence factors [82].

Tissue tropism may be also a driver of super shedder strain evolution, as revealed by phenotypic observations of adherence capabilities specific to host cell lines [81]. In fact, the super shedder strains considered in this study presented with a distinctive adherent aggregative phenotype on recto–anal junction epithelial cells. This tissue is known as the primary colonization site of *E. coli* O157:H7.

Host specificity may be another driver for the evolution of super shedder *E. coli* O157:H7 strains. By comparing two *E. coli* O157:H7 super shedder strains, Katani et al. [80] observed a signature of positive selection in the genes involved in the virulence and adaptation with a host. Teng et al. [83] found that the most divergent gene categories were functionally associated with host specificity and environmental interactions.

Lastly, it has been suggested that the occurrence of super shedding events may be related to the recurrent formation and sloughing of biofilms including the pathogen [39,84]. This hypothesis is in line with comparative studies showing evidence of a strong adherent aggregative phenotype of *E. coli* O157:H7 super shedder strains, in particular adhesion to the epithelial cells of the bovine recto–anal junction [81]. Reversible attachment to a surface is a prerequisite for biofilm formation [39]. According to Castro et al. [84], the biofilm hypothesis may also explain a correlation previously observed between super shedding and low water consumption in bovine hosts, as a low feces hydration may cause friction in the recto–anal junction and lead to an increased detachment of the biofilms. Nevertheless, genomic comparisons either revealed no differences in the sequence of the genes involved in biofilm formation [79] or suggested a possible reduced biofilm formation capability (e.g., caused by a truncation of the *cah* gene [81]). However, cyclic biofilm sloughing and formation do not explain why super shedding may remain persistent; the alternative is that the formation and sloughing of multiple biofilms occur simultaneously. It has also been suggested that the biofilm may be related to a long-lasting survival of *Salmonella*, which, in turn, may cause a prolonged shedding [85]. This hypothesis about the role of biofilms raises questions concerning their initial conditions of formation, in particular how the pathogen may first survive in the lumen and then access the surface of epithelial cells. A dysbiotic context including a permissive microbiota filter and an altered mucus layer might be a prerequisite for biofilm formation.

Taken together, the abovementioned studies highlight that super shedding partially relies on strain-levels differences, although it is a complex phenomenon including host-related factors.

### 4.4. Host Related Factors: Age and Sex

Studies of risk factor associated with super shedding have demonstrated relationships with host age. Early-lactation heifers are more likely to shed *E. coli* O157:H7 than older cows [57,60]. According to the authors of these studies, the metabolic challenge caused by lactation may explain why the host may be more prone to become a super shedder. In their meta-analysis of *M. avium* subsp. *paratuberculosis* shedding by experimentally infected calves and cows, Mitchell et al. [86] also found a dependency of the shedding patterns on the age of exposure. Only a few animals shed the pathogen when they reach the lactating age, whereas the majority of the youngest infected calves in this study presented an early shedding phase. The age dependency of the super shedding phenomenon may also rely on the development of immune response and the gut microbiota. Young, immature animals may be more prone to developing a super shedding state. In comparison to age, the sex dependency of super shedding remains unclear. On one hand, some authors suggest a significant relationship. This is the case for the shedding of *E. coli* O157:H7 in a study by Nielsen et al. [84]; in line with this, Chase-Topping et al. [30] found that the presence of female host individuals is related to increased shedding. On the other hand, other studies did not report any relationship between sex and super shedding [87,88]. Possible explanations for this relationship include sex-related differences in the gut microbial composition. However, such differences were mainly observed at the herd level, thus implying possible differences in management practices. For example, herds dedicated to beef production include a higher proportion of calves, these individuals being otherwise sold after birth.

### 4.5. Host Related Factors: Immunity

Intuitively, host immunity should play a key role in the occurrence of shedding phenotypes. A strong immune response should wipe out the pathogen, leading to a low shedder phenotype. In line with this assumption, in *E. coli* O157:H7-shedding cattle, transcriptomic studies of the recto–anal junction revealed reduced expression of genes involved in adaptive and innate immune responses in super shedders [89], as well as a possible immunodeficiency of super shedders at this site of the gastrointestinal tract. Along the whole gastrointestinal tract, Wang et al. [90] observed varying patterns, including increased T-cell migration and proliferation at the distal jejunum and descending colon in *E. coli* O157:H7 super shedders. The patterns of miRNA expression involved in immune functions and lipid metabolism confirmed the differences between the super and non-shedder cattle [89,91], mostly in the recto–anal junction and distal jejunum sections.

The protective role of host immune mechanisms involved in pathogen shedding has been tackled in several other studies. In the case of *Salmonella*, Rogers et al. [64] assigned a crucial role to the inflammatory response with respect to the fate of the pathogen, highlighting that it may lead to a successful engraftment of the pathogen. Intestinal inflammation may cause the secretion of nitrate, tetrathionate, lactate, and oxygen in the gut lumen, which favor the expansion of facultative anaerobes, such as *Salmonella* and other endogenous Enterobacterales, over the strict anaerobic bacteria. During the mucosal inflammatory response, the competition taking place within the microbiota may lead to a luminal expansion of *Salmonella*, suggesting that the super shedder host elicits a strong intestinal inflammation, favoring colonization by *Salmonella*.

This hypothesis is in line with several works performed in different animals showing a strong inflammation in super shedder mice and pigs [7,34,35,36]. These studies demonstrated that low shedders tended to respond faster than super shedders [36], with a quicker inflammatory response [35]. Among the differentially expressed genes, Huang et al. [34] reported an overexpression of genes associated with the IFN-γ signaling pathway in super shedders (which was in line with results reported by Uthe et al. [92] showing an increased level of circulating IFN-γ in super shedders) and an overexpression of other regulators, such as TLR4, CEBPB, and SPI1, in super shedders. Knetter et al. [35] found that the STAT1, IFNB1, and IFN-γ networks were upregulated in super shedder pigs, whereas genes negatively regulating the immune response were found to be overexpressed in low shedder pigs. This was associated with increased serum IFN-γ, IL-1β, and TNF-α compared with low shedding pigs and reduced CXCL8 in super shedder pigs. Using a WGCNA (weighted gene co-expression network analysis) approach, Kommadath et al. [36] described several groups of genes showing expression patterns associated with *Salmonella* shedding, with the patterns of expression partially consistent with those reported by Huang et al. [34].

A role of the inflammatory response may be also suggested by the existence of resistant and sensible lineages, showing differences in terms of the level of *Salmonella* carriage. A study by Cazals et al. [93] shed light on the link between lineage resistance, heterogeneous shedding patterns, and gut microbial composition (including anti-inflammatory features). The authors found that the production of anti-inflammatory short-chain fatty acids may have been enriched in the resistant chicken under scrutiny and in the low carriers of the more susceptible line.

Innate immune responses were traditionally thought to be non-specific and without the capacity to acquire a memory phenotype. Emerging evidence shows that innate immunity exhibits an immunological memory against past insults known as trained immunity [94]. Trained immunity confers broad-spectrum protection against several lethal infections. In the context of the super shedding status, the role of innate training is not known, but if it has the capacity to reduce shedding of the organism, its use may provide significant benefit to the health of the herd by reducing the risk of disease transmission. The impact of innate training on tissue pathology, pathogen burden, and overall disease outcome needs to be carefully evaluated in the context of particular disease settings in order to determine the risk vs. reward of engaging innate memory [95].

Taken together, protective mechanisms during infection and reinfection cannot be solely attributed to adaptive immune responses but also to innate immune responses and to the microbiota-nourishing immunity linked to colonization resistance [96].

### 4.6. The Interplay between the Host, the Pathogen, and the Gut Microbiota

As no single factor satisfactorily summarized the super shedding phenomenon, it may emerge as a result of an interaction between multiple factors (see also [64]), including factors related to the host and its gut microbiota and to the pathogen itself. Under this scheme, low shedding may arise when a strain faces strong competition from the gut microbiota both before and after the inflammatory response (Figure 2). In contrast, a super shedder strain creates a suitable environment in the host thanks to the virulence factors, which highjack the inflammatory response and overcome the barrier effect conferred by the gut microbiota (Figure 3).

## 5. Control Strategies

### 5.1. Husbandry Practices

Control strategies against the pathogens responsible for the super shedder phenomenon include several biosecurity measures that may lead to a reduction in shedding, even if they do not target the heterogeneity of infection per se.

For instance, water and feed contamination have been mentioned as a potential issue with respect to the control of pathogen shedding [97,98]. Depending on the host and pathogen, controlling for water and feed contamination may nevertheless achieve contrasting results. In a recent systematic review, Pessoa et al. [99] demonstrated that biosecurity measures including those targeting water and feed contaminations might be inefficient in the case of *Salmonella* in chickens. In line with this, water disinfection and feed hygiene may also only slightly affect the epidemiology of *E. coli* O157:H7 in cattle [97]. In contrast, in a review focusing on biosafety measures in pig production, Rodrigues da Costa et al. [100] found that water treatment (in a broad sense) might be efficient against *Salmonella*, although it does not appear to be the most effective intervention. Hygienic actions in drinking water may also be efficient against *Campylobacter* spp. in chickens [99].

Improving the hygiene of livestock housing may also aid in pathogen control [97,98]. Pessoa et al. [99] reported that high-efficiency cleaning and disinfection measures were associated with successful control of *Salmonella* and *Campylobacter* in chickens. This is also the case for *Salmonella* in pigs [100]. In line with this, Ellis-Iversen et al. [101] demonstrated that a package of measures aimed at maintaining a clean environment and closed groups of young stock significantly reduced the burden of *E. coli* O157:H7. Similarly, testing and culling may be an appropriate means of controlling *M. avium* subsp. *paratuberculosis* [102], and if combined with good management practices (i.e., cleaning calving pens between usage and individual housing of calves), it may be a sufficient measure to control super shedders and subsequent pathogen transmission.

Studies conducted by De Cort et al. [98] indicated that stressful flock conditions may exacerbate *Salmonella* shedding in layer chickens, in particular at the time of molting. Alternative molting induction methods may thus help to control shedding. This is in line with a modelling study showing that reduced stress may be crucial in keeping *Salmonella* prevalence in pigs as low as possible [103]. Similar phenomena have been reported for other pathogens [104].

Although difficult, the exclusion of wildlife reservoirs, including insects, rodents, and birds, has also been suggested as a potential means to reduce infection. Subsequently, this may facilitate the reduction in pathogen shedding, specifically in relation to *Salmonella* in chickens [98] and *E. coli* O157:H7 in cattle [97]. However, there is a paucity of data related to super shedding in wild animals; therefore, this should be an area of focus for future research.

### 5.2. Vaccines

Immunization strategies against pathogen infection have been shown to be effective against shedding, although they do not specifically target the phenotype of super shedder animals. An ideal vaccine should facilitate the control of all facets of a disease (e.g., clinical forms, tissue colonization, and pathogen shedding) [105]. However, this is not always possible. For instance, in pigs, it is not possible to completely prevent *Salmonella* colonization during the production cycle. *Salmonella* can be found in dams and early on in environmental samples. Therefore, in this context, the vaccination is used to control clinical forms and subclinical shedding [106]. This is also the case for *Salmonella* in chickens and *E. coli* O157:H7 or *M. avium* subsp. *paratuberculosis* in cattle, for which vaccination may be used to decrease pathogen shedding levels rather than to control the associated disease [24,105,107].

An issue associated with this control strategy is cross protection. For example, a vaccine against *S.* Typhimurium strains poorly protects animals against *S.* Enteritidis. In the case of *Salmonella*, reducing shedding through vaccination implies a wide variety of serovars that must be taken into account [106]. Obtaining such cross protection among the serovar is a difficult task. In line with this, cross protection against Shiga toxin-producing *E. coli* may also be an issue in the case of vaccines targeting *E. coli* O157:H7 [24].

An effect on pathogen shedding may be observed with different kinds of vaccines. Commonly used vaccines include live attenuated and inactivated vaccines. Inactivated vaccines are likely to be administered via the parenteral route and induce a local and systemic response, although this response may not be long-lasting. Such vaccines are typically well-tolerated and lack the risk of reversion and subsequent infection. They are also stable; therefore, it is not usually necessary to keep them in a cool chain. Otherwise, live attenuated vaccines are likely to induce a long-lasting local and systemic response, in addition to inducing effects on the local microbiota [108].

### 5.3. Bacteriophages

Bacteriophages can be utilized at all stages of animal production to reduce pathogenic bacterial loads [109]. Phages are highly specific and usually only infect a single bacterial species or even specific strains within a species. Thus, phage addition would not have an considerable impact on the gut microbiota. When they are used for preharvest control as therapeutic strategy, they may help to reduce pathogen colonization, persistence, and shedding.

Some commercial phage products may achieve limited efficiency in the reduction in shedding (e.g., [110]). Nevertheless, many studies have reported encouraging results. Tanji et al. [111] inoculated a three-phage cocktail (SP21, SP22, SP15; 10^10^ PFU) on a daily basis in mice previously infected with 10^9^ CFU of *E. coli* O157:H7. Under this framework, they observed that the levels of colonizing bacteria in the gastrointestinal tract fell below 10^1^ CFU/g. Callaway et al. [112] used an eight-phage cocktail in a sheep model targeting *E. coli* O157:H7 and observed that shedding was reduced from 10^7^ to 10^4^ CFU/g. It was demonstrated that the use of a suspension of phages specific to *C. jejuni* and *C. coli* in the water or feed of broiler chickens caused a significant decrease of nearly 2 log10 CFU/g in colonization by both species of bacteria over 7 days [113]. Using a chicken model and phages targeting *Salmonella*, Bardina et al. [114] observed a reduction in *Salmonella* cecal colonization, ranging from 4.4log_10_ CFU/g to 2log_10_ CFU/g, depending of the schedule of phage inoculation (10^10^ PFU each).

Nevertheless, in the latter study, a long-term reduction in *Salmonella* colonization was only observed after frequent phage treatment, which calls into question its practical implementation as treatment in commercial broilers. Another issue is the host range of the phage product. Ideally, it should be wide enough to kill members of the target species. Lastly, the natural capability of bacteria to develop phage resistance may be another point of concern [115]. The use of broad-spectrum phage cocktails may help to address this problem [109].

### 5.4. Feed Additives

Antipathogen strategies include interventions with respect to animal diets. Historically, owing to the availability of cheap antibiotics, such interventions were of limited effectiveness. This has changed since the emergence of antibiotic resistances have raised growing concern [116]. Feed additives include probiotics that are live, non-pathogenic bacteria used as direct-fed microbials to improve the health and gut equilibrium. They may strengthen the intestinal immune response and barrier function [117]. Thus, they can be used to control pathogens [118] and even to reduce fecal shedding.

This has been shown in the case of weaned ruminants infected with *E. coli* O157:H7; for example, using a combination of 18 strains able to inhibit the growth in vitro of *E. coli* O157:H7 (17 *E. coli* and 1 *Proteus mirabilis*), Zhao et al. [119] demonstrated a reduction in fecal shedding. After a challenge with 10^10^ CFU following inoculation with the probiotic mixture, no bacteria were recovered from five of the six probiotically treated animals. This was observed at several time points until 28 days. In contrast, *E. coli* O157:H7 was shed by all of the control animals throughout the experiment. This result is in line with those of reported by Lema et al. [120], who used a lamb model and several combinations of *Lactobacillus* and *Streptococcus* probiotic strains, and a challenge of 10^10^ CFU; under this scheme, the authors observed a maximal reduction of 3.3 log_10_ CFU/g feces of shed bacteria. The potential of several probiotic mixtures has also been tested in field studies ([121,122]; see [123] for a review). Many of these works are based on combinations including *Lactobacillus* spp. strains [123,124].

The beneficial effect of probiotics on *Salmonella* shedding has also been demonstrated. Casey et al. [125] used a five-strain probiotic combination in a pig model. Following a 6-day treatment with this mixture, the animals were challenged with an *S.* Typhimurium strain. The authors observed a significant reduction in *Salmonella* fecal shedding in the probiotically treated group (up to 2.28 log_10_ CFU/g feces on day 15 after infection). Studies showing a reduction in cecal colonization after a probiotic treatment also suggest an influence on fecal shedding. For example, significant differences in colonization of the cecal content were reported found by Carter et al. [126] in chickens infected with *S. Enteritidis* after inoculation with a probiotic mix.

In line with the use of probiotics in adult hosts, some studies argue for the use of well-defined mixes of bacteria in young animals to influence later host–pathogen interactions. For instance, it has been demonstrated that a specifically designed bacterial community may influence the maturation of the host’s immune system [127]. In a germ-free mouse model, it was shown that a defined bacterial mix may lead to colonization resistance [128]. In another chicken model, Kempf et al. [25] demonstrated that inoculating young chicks with a mix of four commensal strains before infection may lead to a reduction in *S.* Enteritidis shedding (up to 2 log_10_ CFU/g feces two weeks after infection) and, above all, complete abolition of the presence of super shedder chicks (Kempf et al. personal data). Besides potential effects on the host’s immune response, these results suggest that probiotic products may modulate the gut microbiota to induce antipathogen features. Consistent with this idea, Brugiroux et al. [128] observed that the inoculated minimal defined microbiota was stable over consecutive mouse generations and provided partial colonization resistance to *Salmonella*. Thanks to predictions made by comparative metagenome analyses, the authors next developed an improved mix able to provide a colonization resistance comparable to that of conventional complex microbiota. These results suggest that defined mixes can be used as starting microbiota to control for pathogen shedding.

Other feed additives have shown their efficiency against pathogens, including prebiotics, defined as non-digestible food ingredients that favor several enteric bacteria, therefore improving host health. For instance, this has been shown experimentally for sodium chlorate addition in the case of *E. coli* O157:H7 with a shedding drop exceeding 2 log CFU/g [129]. Other products, include carbohydrates, have shown partial efficiency against pathogen shedding [130,131]. Again, the underlying mechanisms may include a modulation of the gut microbiota [132].

## 6. Concluding Remarks

Heterogeneity of shedding patterns is of considerable concern with respect to disease, control especially as it is mainly super shedders that transmit pathogens to congeners and then to slaughterhouses. However, little is known about the development of a super shedding phenotype. A standard definition of the super shedder category remains problematic (Figure 1). Several methods have been proposed in a few well-studied models of enteric pathogens. In particular, it is necessary to differentiate between transient and persistent super shedding, with the latter corresponding to distinct individuals characterized by particular host–pathogen–gut microbiota relationships. Beyond definitions based on shedding patterns, growing knowledge of host and microbiota factors facilitate the definition of markers indicative of the super shedding state in the near future, including diagnostic and predictive markers. These markers are expected to play an important role in strategies for the control or reversal of super shedder-susceptible phenotypes for predictive markers. This should be a crucial asset, alongside the current methods based on husbandry practices, vaccines, bacteriophages, and feed additives. However, we are far from a consensus about the general mechanisms of super shedding; besides neglected questions (the possible role of coinfections, sex dependency, differences among host species, and host- and microbiota-derived metabolites), different combinations of host, pathogen, and gut microbiota features may lead to a similar outcome, even in the same model. This highlights the need for large-scale functional studies reproducing this phenomenon under controlled conditions.

## Figures and Tables

**Figure 1 microorganisms-10-02101-f001:**
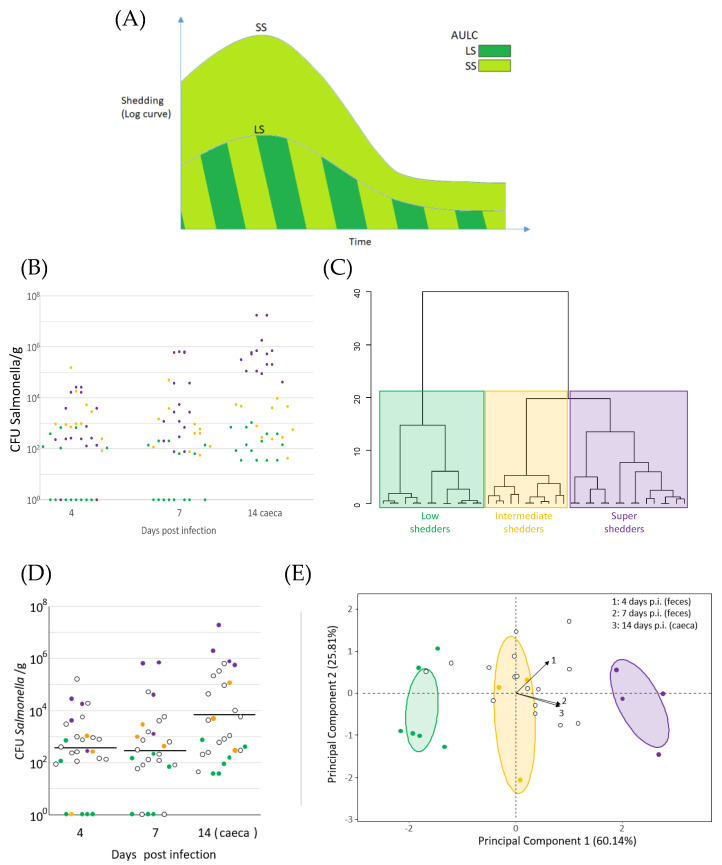
Definitions of the super shedder category taking into account the dynamic of infection (**A**) scheme representing the areas under the log curves (AULC) describing the shedding levels of super shedders (SS) and low shedders (LS). The fecal counts at different time points are log−transformed, and the cumulative area under the plotted log curve (AULC) is determined [32,34,35,36]. (**B**) Shedding levels of *Salmonella* super shedders (purple), as well as intermediate (orange) and low shedders (green) in chickens; data adapted from [33]. (**C**) Hierarchical clustering of individuals based on these shedding levels; data adapted from [33]. (**D**) Shedding levels of *Salmonella* super shedders (purple), as well as intermediate (orange) and low shedders (green) in chickens [25]. (**E**) PCA summarizing these shedding levels; adapted from [25].

**Figure 2 microorganisms-10-02101-f002:**
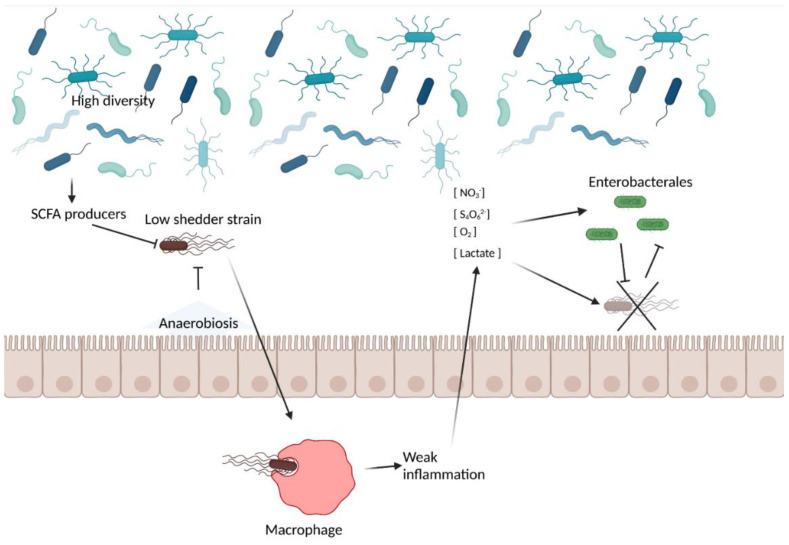
Possible mechanisms at play in the low shedder phenotype. A gut with a high diversity and a high microbiota-nourishing immunity, via the presence of bacteria conferring a strong barrier effect, such SCFA producers, that promote epithelial anaerobiosis, may prevent the installation of the pathogen. This may be exacerbated by the lack of intrinsic virulence factors, i.e., if the pathogenic strain itself is more prone to generate a low shedder phenotype. If the pathogen succeeds in eliciting an inflammatory response required to create a new niche for pathogen installation, it may outcompete the resident gut microbiota in strong competition with closely related microbial bacteria (e.g., Enterobacterales in the case of *Salmonella* and *E. coli* O157:H7), which may prevent its luminal expansion. Created with BioRender.com, accessed on 18 October 2022.

**Figure 3 microorganisms-10-02101-f003:**
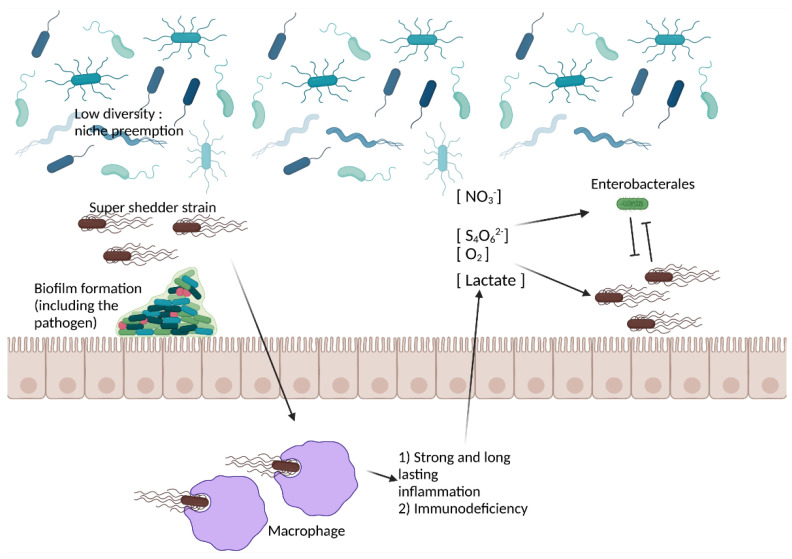
Possible mechanisms at play in the super shedder phenotype. A gut presenting with low diversity exhibits low microbiota-nourishing immunity and may be more prone colonization by the pathogen. Virulence factors may induce a strong inflammatory response, thus creating a new niche for pathogen installation, altering its metabolism. Then, the pathogen faces competition from other closely related bacteria (e.g., Enterobacterales in the case of *Salmonella* and *E. coli* O157:H7), thanks to specific bacterial compounds, which may lead to its luminal expansion. Alternatively, immunodeficiency may lead to luminal expansion of the pathogen. Created with BioRender.com, accessed on 18 October 2022.

## Data Availability

Not applicable.

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
