# Peer review of "Super Shedding in Enteric Pathogens: A Review"

_microorganisms, 2022, doi:10.3390/microorganisms10112101_

Round 1

Reviewer 1 Report

The review by Kempf et al. is well organized and updated but writing style has scope for improvement like more concise, however, microbiome and enteric bacterial correlation is poorly explained and it has a high potential to improve. How diet, enteric pathogens, age and sex is important for super shedding is missing. Another important aspect that significantly improve this review is to include how tropism and persistence mechanism is important for super shedding. 

Author Response

The authors would like to thank the reviewers for their work in reviewing our article entitled “Super-shedding in enteric pathogens: a review”. Their remarks and comments were very relevant. We have taken into account the majority of the Rewiever’s comments and we have provided a point-by-point response below.

The line numbers refer to the manuscript version preceding the revision.

Reviewer 1=======================================

The review by Kempf et al. is well organized and updated but writing style has scope for improvement like more concise, however, microbiome and enteric bacterial correlation is poorly explained and it has a high potential to improve.

According to the Reviewer 1’s comment, the article has been re-read and we have tried to be more concise. We do not agree with the comment “microbiome and enteric bacterial correlation is poorly explained”. In the section entitled “Gut microbiota-related factors” we investigated two hypotheses about the super shedding, considering both the level of the microbiota and the level of particular enteric bacteria. These hypotheses are (1) a possible relationship between the microbiome α-diversity and super-shedding phenotype (2) possible correlations between particular enteric bacteria and super-shedding phenotype, suggesting direct or indirect effects on the pathogen (lines 310-361). The lack of a consensus about these relationships, the potential roles of others actors of the microbiome are also evoked below (lines 376-403).

How diet, enteric pathogens, age and sex is important for super shedding is missing.

  • The relationship between diet composition and pathogen shedding is mentioned on line 294. We have added the following sentence: “The role of diet composition, in particular, has been extensively reviewed elsewhere (e.g. Jacob et al. 2009)”. Dietary interventions are presented in the last section (lines 670-725).
  • By mentioning “enteric pathogens”, Reviewer 1 should refer to coinfections (i.e. how a coinfecting pathogen may affect the shedding of another given pathogen). In fact, we did not find references clearly stating that this may be an important factor for the super shedding of bacterial enteric pathogens. We have nevertheless added a concluding paragraph including a sentence that evoke the possible role of coinfections (below the line 725).
  • Age is indeed a major host-related factor of super shedding and this point is presented in the paragraph entitled “Age” on lines 469-481.
  • Given the lack of studies about the sex effect (except a few ones focusing on E. coli O157:H7), we have added a sentence in the concluding paragraph mentioning sex-dependency as a possible perspective for further researches (below the line 725). We have also added a paragraph on the relationship between sex and super shedding (the “Age” section being now entitled “Age and sex”):

“By comparison to the age, the sex-dependency of the super shedding remains unclear. On one side, some authors point out a significant relationship. This is in case for the shedding of E. coli O157:H7 in the study of Nielsen et al. [84]; in line with this, Chase-Topping et al.  [30] also found that a presence of female host individuals is related to a higher shedding. On the other side, other studies did not report any relationship between sex and super shedding [84, 85]. Possible explanation of this relationship include sex-related differences of the gut microbial composition. Nevertheless, the differences were mainly observed at the herd level: they may thus imply differences in management practices. For example, herd dedicated to beef production include a higher proportion of calves, these individuals being otherwise sold after birth.”

Another important aspect that significantly improve this review is to include how tropism and persistence mechanism is important for super shedding.

  • We do not understand what the Reviewer means about tropism. We have cited some references presenting the host tissue tropism of super shedder strain (on lines 456-460). Moreover, as explained in the introduction, host tropism in the broad sense seems to lie beyond the scope of the review: “Super shedding appears as a particular feature of heterogeneous infection, insofar as it differs from, as an example, host-specific heterogeneous shedding patterns” (lines 33-34). For the sake of clarity, this has been change into : “Super shedding appears as a particular feature of heterogeneous infection, in that as it should be distinguished from shedding patterns differing among host species”.
  • We agree with Reviewer 1 that pathogen persistence is a condition of the super shedding, as well as any mechanism contributing to the success of infection. Nevertheless, we preferred the paper to focus more on the shedding itself, than on these preliminary steps. For the sake of clarity, we have modified the following sentence of the introduction (lines 34-37): “Indeed, super shedding occurs when a small number of a given host shed high levels of a pathogen, as a result of a successful infection (which imply successful survival, colonization and persistence).”

We hope that with the improvements we have made to this manuscript you will find it suitable for publication.

With our best regards,

Florent Kempf, on behalf of the authors

Reviewer 2 Report

The manuscript presents a comprehensive overview of the phenomenon of bacterial pathogen super shedding in animals. Its structure and style make it easy to follow the subtopics proposed by the authors.

Minor remarks

In the section “Gut-microbiota related factors” the authors do not comment on the methodological aspects (if there are any, i.e. different methodological approaches) that could impact the final alfa-diversity parameter outcome.  

A short paragraph summarising this review or indicating the future directions of bacterial super shading is missing. 

Lines 245-248 the sentence should not be in italics

Line 521 please explain the “WGCNA” abbreviation

Line 647 “would not”

Line 657 “that the use”

Author Response

The authors would like to thank the reviewers for their work in reviewing our article entitled “Super-shedding in enteric pathogens: a review”. Their remarks and comments were very relevant. We have taken into account the majority of the Rewiever’s comments and we have provided a point-by-point response below.

The line numbers refer to the manuscript version preceding the revision.

=Reviewer 2==========================================

The manuscript presents a comprehensive overview of the phenomenon of bacterial pathogen super shedding in animals. Its structure and style make it easy to follow the subtopics proposed by the authors.

Minor remarks

In the section “Gut-microbiota related factors” the authors do not comment on the methodological aspects (if there are any, i.e. different methodological approaches) that could impact the final alfa-diversity parameter outcome. 

We agree with Reviewer 2 that there may be here interpretation biases about the α-diversity outcome. Throughout this section (on lines 310-332) high diversities were associated to high α-diversity values, although this is not a general rule: for example, the Simpson index ranges from 0 to 1, with 0 representing infinite diversity and 1 representing no diversity. Nevertheless, all references cited in this section are based on index correlating high values and high diversity, like the Shannon index. For the sake of clarity, we have the modified the section on lines 311-313 into : “This suggests that the super shedder phenotype should be linked with a lower α-diversity, whereas the low-shedder one should be linked with a higher α-diversity (assuming that one use an α-diversity index correlating high values and high diversity; e.g. Shannon index)”.

A short paragraph summarising this review or indicating the future directions of bacterial super shading is missing.

We have added the following paragraph at the end of the manuscript, below the line 725 :

Concluding remarks

Heterogeneity of shedding patterns is of great concern for disease control especially as it is mainly the super shedders that transmit pathogens to congeners and then to slaughterhouses. However, little is known about the development of a super shedding phenotype. A standard definition of the super shedder category remains problematic (Figure 1). Several methods have been purposed in a few well-studied models of enteric pathogens. In particular, it is necessary to differentiate between transient and persistent super shedding, the latter corresponding to distinct individuals characterized by particular host-pathogen-gut microbiota relationships. Beyond definitions based on shedding patterns, growing knowledge of host and microbiota factors will allow in a near future to define markers indicative of the super shedding state, including diagnostic and predictive markers. These markers are expected to play an important role in strategies for the control or reversal of super shedder-susceptible phenotypes for predictive markers. This should be a crucial asset, alongside the current methods based on husbandry practices, vaccines, bacteriophages or feed additives. We are nevertheless far from a consensus about the general mechanisms of super shedding: beside neglected questions (possible role of coinfections, sex-dependency, differences among host species, host- and microbiota-derived metabolites), different combinations of host, pathogen and gut microbiota features may lead to a similar outcome even in the same model. This raise the need for large-scale functional studies reproducing the phenomenon under controlled conditions.“ 

Lines 245-248 the sentence should not be in italics; Line 521 please explain the “WGCNA” abbreviation; Line 647 “would not”; Line 657 “that the use”

These remarks have been taken into account.

We hope that with the improvements we have made to this manuscript you will find it suitable for publication.

With our best regards,

Florent Kempf, on behalf of the authors

Reviewer 3 Report

Comments to the authors

This review is a very nice and important review and for me introduce an interesting information.  There some points need clarification from the authors:

-          How we investigate and detect the Super shedders?

-          Temporal shedding patterns: is the super-shedder among them?

-          Standards should be design to judgment and detect super-shedders.

-          the carrier host individuals may temporally become Salmonella super shedders; how?

-          Where are the viruses?

-          Where are protozoa like Cryptosporidium, Giardia and Entaoemba?

-          Engraftment? What means? In line 310

-          Are the infectious diseases have a role in the supper-shedder phenomenon? In another words for example; infection by a GIT parasite may be a role in this!

-          Control strategies lost antibiotic, essential oils and pro & pre biotics.

Author Response

The authors would like to thank the reviewers for their work in reviewing our article entitled “Super-shedding in enteric pathogens: a review”. Their remarks and comments were very relevant. We have taken into account the majority of the Rewiever’s comments and we have provided a point-by-point response below.

The line numbers refer to the manuscript version preceding the revision.

This review is a very nice and important review and for me introduce an interesting information.  There some points need clarification from the authors:

  • How we investigate and detect the Super shedders?

Some facets of the problems are tackled in the paragraph entitled “Delineation of the super shedding category”. Nevertheless, we mainly focused on the analysis of the shedding patterns. We have added the following sentence: “The delineation of super shedder category stems from the characterization of the shedding patterns, which can be assessed using several quantitative methods (in practice, most of studies are based on pathogen counts) and sampling method (swabbing, collection of caecal or faecal material from dead or living hosts)”.

  • Temporal shedding patterns: is the super-shedder among them?

Reviewer 3 should refer to the puzzling cases of intermittent shedding (in other words ‘transient super shedding’). We actually raised this question in the manuscript (lines 111-114 : “Because of these contrasted individual shedding patterns, some authors have purposed that the super shedding does not form a distinct category of individuals, but it should be considered as a transient property that appear and disappear over time [Munns et al. 2014, Spencer et al. 2015]”). Nevertheless, according to other studies, we emphasized that the super shedders may also constitute a distinct, persistent category characterized by their own mechanisms. For the sake of clarity, we have added a concluding paragraph (below the line 725) including a sentence on the link between transient and persistent super shedding.

  • Standards should be design to judgment and detect super-shedders.

We agree with Reviewer 3 that a standard definition of the super-shedding might be a first step to investigate the phenomenon. For this reason, the different methods found in the literature were summarized in the manuscript (lines 126-169, Figure 1). We have also added a concluding paragraph including a sentence on the interest of a standard definition.

  • the carrier host individuals may temporally become Salmonella super shedders; how?

As explained above, in the present review we mainly focused on the super shedding taken as a distinct category of individuals presenting persistent high shedding patterns. Intermittent shedding – when it is not an artefact of the sampling method - is a distinct phenomenon, whose mechanisms are largely unknown. In particular, a better understanding of this phenomenon should imply specific models where the shift between the different states may be easily reproduced. To clarify the distinction between transient and persistent super shedding, we have added a concluding paragraph including a sentence on the link between transient and persistent super shedding.

  • Where are the viruses? Where are protozoa like Cryptosporidium, Giardia and Entaoemba?

As it would be too cumbersome to describe all super shedding phenomena in bacteria, viruses and parasites, this review focuses on the super shedding of enteric bacteria. However, the super shedding phenomena has been well described with viruses (which is mentioned on lines 27-30). Besides we did not found numerous articles describing the super shedding in parasitic models, including the parasites mentioned by Reviewer 3.

  • Engraftment? What means? In line 310

This has been changed into “colonization”.

  • Are the infectious diseases have a role in the supper-shedder phenomenon? In another words for example; infection by a GIT parasite may be a role in this!

Reviewer 3 refers to the fact that coinfections/multi species infection may alter the shedding of a given pathogen and thus favor the emergence of super shedders (e.g. Lass et al. 2013, Nguyen et al 2021). To our knowledge, this has been never observed for enteric pathogenic bacteria. We have nevertheless added a concluding paragraph including a sentence that evoke the possible role of coinfections.

Lass, Sandra, Peter J. Hudson, Juilee Thakar, Jasmina Saric, Eric Harvill, Réka Albert, and Sarah E. Perkins. "Generating super-shedders: co-infection increases bacterial load and egg production of a gastrointestinal helminth." Journal of the Royal Society Interface 10, no. 80 (2013): 20120588.

Nguyen, Nhat, Ashutosh K. Pathak, and Isabella M. Cattadori. "Gastrointestinal Helminths Increase Bordetella bronchiseptica Shedding and Host Variation in Supershedding." bioRxiv (2021).

  • Control strategies lost antibiotic, essential oils and pro & pre biotics.

We agree with Reviewer 3 that such strategies may target a reduction of the bacterial fecal shedding. The current manuscript already includes a paragraph on the use of probiotics (lines 670-699) and prebiotics (line 718-725). By comparison, only few studies focus on antibiotics and essential oils, in particular large-scale in vivo studies using it as dietary additives. Among them, we did not find references showing a relevant effect on bacterial shedding. For example, Alali et al. (2013) did not observe effects of their essential oil compounds; Allali et al. (2004) did not observe effect of an antibiotic supplementation (using oxytetracycline and neomycin).

Alali, W. Q., C. L. Hofacre, G. F. Mathis, and G. Faltys. "Effect of essential oil compound on shedding and colonization of Salmonella enterica serovar Heidelberg in broilers." Poultry science 92, no. 3 (2013): 836-841.

Alali, W. Q., J. M. Sargeant, T. G. Nagaraja, and B. M. DeBey. "Effect of antibiotics in milk replacer on fecal shedding of Escherichia coli O157: H7 in calves." Journal of animal science 82, no. 7 (2004): 2148-2152.

We hope that with the improvements we have made to this manuscript you will find it suitable for publication.

With our best regards,

Florent Kempf, on behalf of all authors